# Endophyte *Bacillus subtilis* evade plant defense by producing lantibiotic subtilomycin to mask self-produced flagellin

Yun Deng [1], Hanqiao Chen[1], Congzhi Li[1], Jianyi Xu[1], Qingdong Qi[1], Yuanyuan Xu[1], Yiguang Zhu[1], Jinshui Zheng [1], Donghai Peng[1], Lifang Ruan[1] & Ming Sun[1]*

Microbes can enter into healthy plants as endophytes and confer beneficial functions. The entry of commensal microbes into plants involves penetrating plant defense. Most mechanisms about overcoming plant defense are focused on adapted pathogens, while the mechanism involved in beneficial endophyte evades plant defense to achieve harmonious commensalism is unclear. Here, we discover a mechanism that an endophyte bacterium *Bacillus subtilis* BSn5 reduce to stimulate the plant defensive response by producing lantibiotic subtilomycin to bind self-produced flagellin. Subtilomycin bind with flagellin and affect flg22-induced plant defense, by which means promotes the endophytic colonization in *A. thaliana*. Subtilomycin also promotes the BSn5 colonization in a distinct plant, *Amorphophallus konjac*, where the BSn5 was isolated. Our investigation shows more independent subtilomycin/-like producers are isolated from distinct plants. Our work unveils a common strategy that is used for bacterial endophytic colonization.

[1] State Key Laboratory of Agricultural Microbiology, Huazhong Agricultural University, 430070 Wuhan, China. *email: m98sun@mail.hzau.edu.cn

Microbes exist as early as 4 billion years ago and so ubiquitously in almost all ecosystems that we can say all the animals and plants are living in the world of microbes[1]. Microbes in plant and plant rhizosphere have been paid more attention as the potential understandings are crucial for agricultural development[2]. In a recent study, the importance of microbes in root for plant growth and health has been highlighted[3].

Endophyte refers to bacteria or fungi that reside internally in plant tissues, which can be isolated from the plant after surface disinfection and cause no negative effects on plant growth[4]. Current investigations based on the isolated sequences of plant microbiota further confirmed the general existence of diverse endophytes in plants[5–8]. Some endophytes promote plant growth by transferring nutrition into available forms for plant, triggering induced systemic resistance, and producing antimicrobial compounds and plant hormones[9]. The beneficial functions depend on stable colonization in plant[10]. However, how endophytes deal with the multiple lines of plant defense to establish the stable commensal relationship with plant has rarely been reported[11,12].

The interplay of microbes and plant defense has been well studied in the pathogen–host binary systems[13]. Plants maintain surveillance of the potential microbial invaders by perceiving microbial-associated molecular patterns (MAMPs)[14]. Flagellin is the best-studied MAMP in bacteria. The conserved 22 amino acids epitope, flg22, in the N-terminal region of the flagellin from pathogen bacteria elicits the defensive response after the recognition by the receptor FLS2 in Arabidopsis thaliana[15,16]. To avoid this defensive response, adapted pathogens evolved various effectors to attenuate the pattern-triggered response. For example, P. syringae secretes an effector, AvrPto, into plant cells to block the flg22-induced defensive response by binding the receptor kinase FLS2[17]. The pathogen and host interactions are described as a Zigzag model[18], which is like an arms race. In an arms race, usually one side would lose at the end.

Different from the fighting to death in pathogen and host relationship, beneficial endophytes usually reach a stable win–win situation with plant. The mechanisms involved in facing the plant defense by beneficial endophytes remain largely misunderstood. A few cases mentioned that beneficial microbes downregulate the expression of the MAMPs[19–21], produce the MAMPs with low-elicit ability[22], or produce some required genes[23] to reduce the stimulation of plant defensive response. A tactful strategy used by an endophyte fungal is that it reduces the β-glucan-triggered defensive response of plant through producing a lectin, FGB1, and binding β-glucan[24]. The mechanism that commensal endophyte bacteria used to modulate plant defense is rarely reported like the recent review stated[12].

Here, we describe a novel strategy that endophyte bacteria B. subtilis enhances its colonization in plant through minimizing the stimulation of the defensive response in A. thaliana by producing lantibiotic subtilomycin to bind self-produced flagellin.

## Results

**Subtilomycin interacts with self-produced flagellin.** Our initial target was to control Amorphophallus konjac soft-rot disease. For that we carried out the aiiA gene transgenic breeding as aiiA gene encoding product could degrade quorum-sensing signals of soft-disease pathogen[25]. An endophyte B. subtilis BSn5 was isolated from the callus culture of A. konjac during the transgenic procedure[26]. As the target to control soft-rot disease, we were interested in the mining of active compounds. An antibacterial protein Apn5 of about 30 kDa in sodium dodecyl sulfate–polyacrylamide gel electrophoresis (SDS–PAGE) was isolated by 30% ammonium sulfate precipitation of BSn5 culture

supernatant (Fig. 1a), and showed activity against Bacillus strains. Through non-denaturing native-PAGE assay on Apn5, a continuous protein signal in lane around a high-molecular-weight range (above 70 kDa) showed inhibition activity in parallel gel activity assay (Fig. 1b and Supplementary Fig. 1). We employed two-dimension SDS–PAGE and mass spectrum to identify the antibacterial protein Apn5. From the result, the most abundant protein spots 3–6 around 30 kDa were identified as flagellin, the monomer subunit of flagellum (Supplementary Fig. 2 and Supplementary Table 1). We thought the main component might be the fragments of flagellum or the polymers of flagellin in native gel assay, which could explain the presentation of a continuous protein signal (Fig. 1b). To determine the origin of the inhibition activity, we knocked out the flagellin-encoding gene hag (homologous to fliC) in BSn5 (Supplementary Fig. 3). The Apn5 analog sample from mutant Δhag remained an identical inhibition activity with Apn5 (Fig. 1c). Through tricine–SDS–PAGE analysis, a band with molecular weight of 3–4 kDa showed inhibition activity (Fig. 1d and Supplementary Fig. 4), and this result linked the activity to our previously identified lantibiotic subtilomycin, which has been determined with an exact molecular weight of 3234.36 Da[27]. The Apn5 analog sample from subtilomycin mutant ΔapnB (an essential gene encoding subtilomycin biosynthetical enzyme for dehydration reaction was inactivated[27]) lost antibacterial activity (Fig. 1c). Therefore, we inferred that the isolated antibacterial protein, Apn5, was actually a complex of flagellin and lantibiotic subtilomycin. To verify that prediction, size-exclusion chromatography (SEC) was used to analyze the composition of Apn5. The collected elution fragments were transferred to SDS–PAGE assay and inhibition activity assay to relatively quantify flagellin and subtilomycin, respectively. The results from SEC showed that most of subtilomycin was co-eluted with flagellin from fragments 4 to 10 that correspond to the molecular size ranging from 150 to 30 kDa (Fig. 1e, and reference to criterion in Supplementary Fig. 5), less was eluted from fragments 18 to 19 that correspond to the molecular size range lower than 15 kDa, and no active sample was eluted before fragment 4, although the peak amount of flagellin was present in fragments 2–4 determined by SDS–PAGE assay, compared with the results from standard molecular weight run (Supplementary Fig. 5). These results showed that most subtilomycin formed a complex with the monomers and oligomers of flagellin (30–150 kDa), very less subtilomycin exists in a free state, and subtilomycin could not bind the polymer state of flagellin, the flagellar filament (>150 kDa).

Furthermore, the interaction of subtilomycin and flagellin in vitro was confirmed by the ligand blot and microscale thermophoresis (MST) analysis. The ligand blot showed that the peptides with molecular sizes of around 9, 12, and 15 kDa, which respectively corresponded to the molecular weight of subtilomycin trimers, tetramers, and pentamers, present a blotting signal with 6× His-tagged Hag protein (Fig. 1f and Supplementary Fig. 6). All exact masses of the oligomers of subtilomycin were detected by LC–MS analysis of purified subtilomycin (Supplementary Fig. 7). The MST analysis showed that an ambiguous binding constant value $K_d$ was given as $18.8 \pm 4.15$ based on the concentration of subtilomycin considering the uncertain amount of subtilomycin oligomers (Fig. 1g). To our knowledge, flagellin was a well-studied MAMP to stimulate plant defense[28,29]; therefore, we wonder if the binding of subtilomycin to flagellin could affect flagellin-induced plant defensive response.

**Subtilomycin suppresses Hag-triggered defense in A. thaliana.** To determine whether the flagellin from a beneficial endophyte B. subtilis induces flg22-dependent defensive response, we

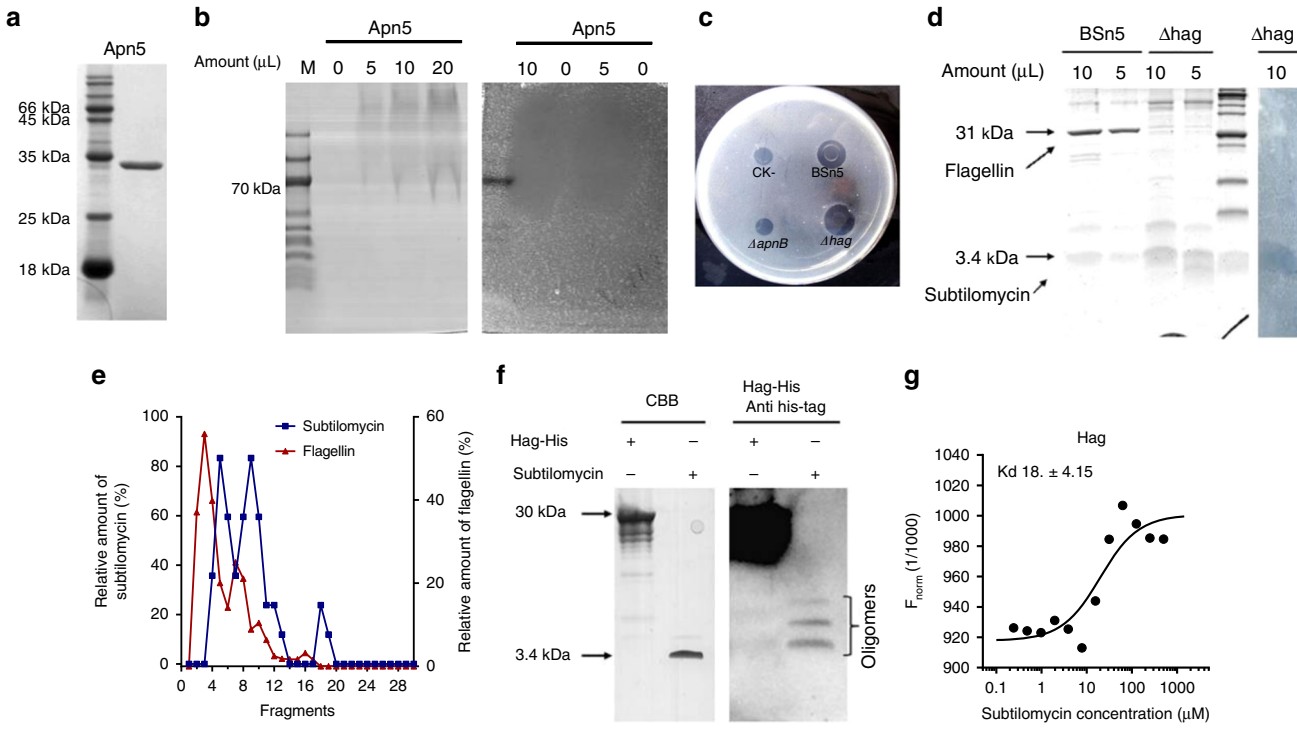

**Fig. 1** The inhibition protein Apn5 is a complex of flagellin and subtilomycin. **a** Denaturing SDS–PAGE analysis of Apn5. **b** Non-denaturing native-PAGE analysis Apn5. In gel inhibition, activity was detected by laying the gel strip on the plate with indicator strain *B. subtilis* CU1065. The loading amount of Apn5 is 5–20 μl, respectively, as the figure presented. **c** Tricine–SDS–PAGE analysis of Apn5 protein from wild-type strain and mutant Δhag; in gel inhibition, activity was detected by laying the gel strip on the plate with indicator strain *B. subtilis* CU1065. **d** Inhibition activity assay on the strain BSn5, mutant *ΔapnB*, and *Δhag* by using CU1065 as an indicator. **e** Size- exclusion chromatography assay of Apn5. The relative amount of subtilomycin in each elution fragment was determined by inhibition activity. The relative amount of flagellin in each elution fragment was determined by SDS–PAGE assay. **f** Ligand blot assay of the purified subtilomycin with Hag-His. CBB coomassie brilliant blue. **g** MST analysis of the interaction of His-tagged Hag and purified subtilomycin. Hag protein was labeled by Cy5-NHS, the final labeled protein concentration is 2 μM, and the series 2× dilutions subtilomycin with the highest final concentration of 500 μM were set as ligand. The dissociation constant value $K_d = 18.8 \pm 4.15$ was calculated by using $K_d$ fit by software NTAnalysis

performed multiple-sequence alignment of flagellin from distinct bacterial species and found that Hag had the conserved N-terminal 22 amino acids epitope, flg22 (Fig. 2a). According to the known structural basis for flg22-induced activation of FLS2–BAK1 complex, the key residue Gly18, which determines the priming of defensive response[30], exists in flg22 (Bs) of Hag (Fig. 2a). However, two residue variations (Ser11Arg and Lys13Gly) compared with flg22 from *Pseudomonas* will lead a decreased ability to elicit defensive response, based on the early site-directed mutagenesis study on flg22 of *P. syringae*[15] (Fig. 2a). By using Hag, mutant ΔHag (flg22) synthesized flg22 (Ps) and flg22 (Bs) as elicitors; we detected the ROS production of wild-type *A. thaliana*, Col-0, and flagellin-insensitive mutant, *fls2-1*, by luminescence assay. The results indicated that Hag induced ROS production in Col-0 but not in *fls2-1* (Fig. 2b). A mutant protein Hag (flg22), in which the flg22 region has been deleted from Hag, did not induce ROS (Fig. 2b). These results indicated that Hag-induced ROS production was dependent on the known flg22–FLS2 signal pathway of *A. thaliana*[30,31], although the inducing level of flg22 (Bs) is much lower than that of the pathogen flg22 (Ps) (Supplementary Fig. 8).

Apn5 is the native form that exists in the supernatant of strain BSn5. We checked if this complex could induce defensive response. We detected the ROS production by luminescence assay. The results showed that the complex Apn5 could not induce the ROS production of *A. thaliana* leaves (Fig. 2c).

To determine whether subtilomycin attenuates the Hag-induced defensive response, oxidative-burst assay was employed.

The results showed that 15 μM subtilomycin significantly depressed Hag-induced ROS production (Fig. 2b, d). The similar results were also observed from the assay of stomatal closure response (Fig. 2e). To verify the suppression effect of subtilomycin in gene level, quantitative real-time PCR and a previously reported dual-luciferase reporter system were used to detect a known flagellin-induced response gene *frk1* (flagellin response kinase 1)[30]. From the results of the expression of gene *frk1*, subtilomycin significantly suppressed Hag-induced *frk1* expression, which is consistent with the ROS production and stomatal closure assay (Fig. 2f, g and Supplementary Fig. 9).

**Subtilomycin binds flagellin at the sites beyond flg22**. To determine whether the defense inhibition effects caused by subtilomycin were dependent on binding with Hag, we applied the synthetic peptides flg22 (Bs) and flg22 (Ps) to test whether subtilomycin suppresses the defensive response induced by these elicitors and interacts with them (Fig. 3a and Supplementary Fig. 10). The results from dual-reporter assays and ROS assays showed that subtilomycin could not relieve defensive response induced by the peptide flg22 (Bs) or flg22 (Ps) (Fig. 3c, d, f, g). MST assay showed that subtilomycin interacted with flg22 (Bs) at a dramatically decreased level ($K_d = 169 \pm 10.4$ μM) and did not interact with flg22 (Ps) (Fig. 3j, k). These results suggest that the defensive response suppression of subtilomycin is more likely dependent on binding flagellin at the region outside the flg22 peptide.

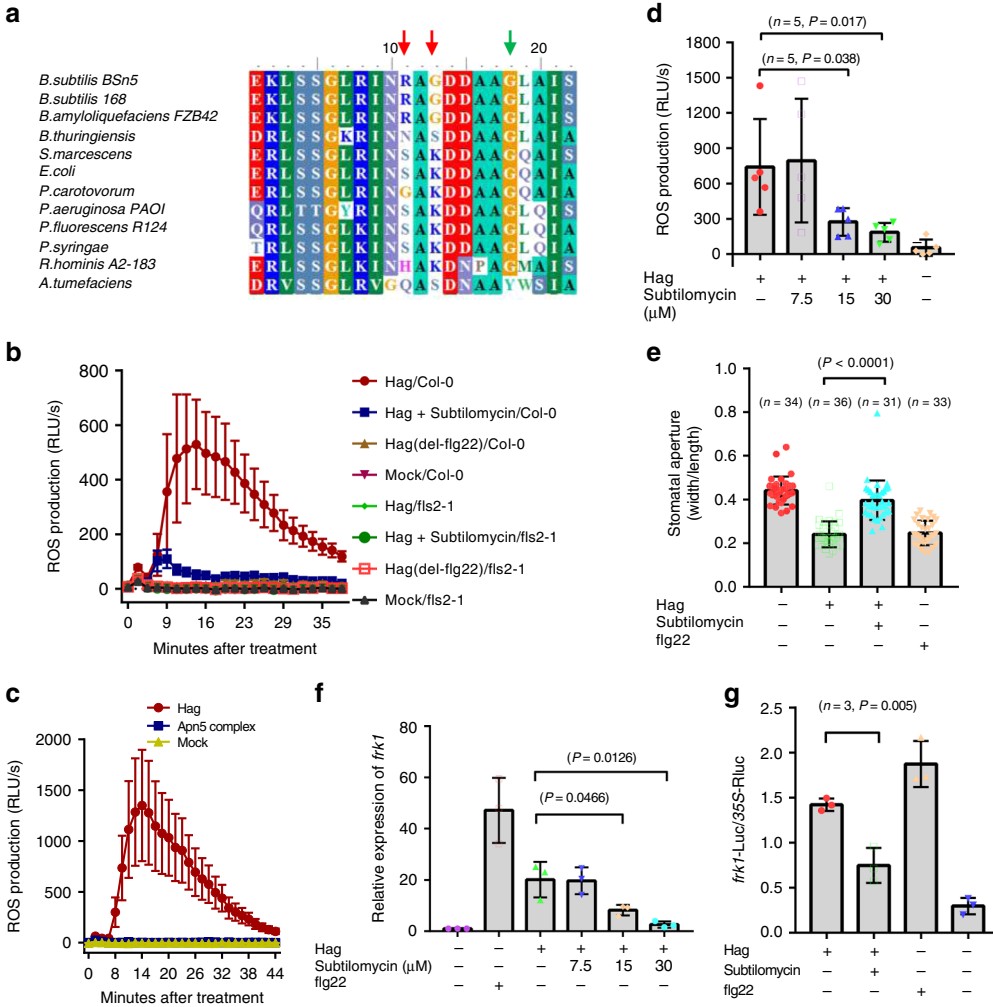

**Fig. 2** Subtilomycin suppresses the Hag-induced defensive response in *A. thaliana*. **a** Sequence alignment of flg22 from different bacteria species. The green arrow indicated that the key residue Gly$_{18}$ was identical with *P. syringae*. The red arrows indicated the two variant residues compared with *P. syringae*. **b** Detection of ROS production in *A. thaliana* Col-0 and *fls2-1* treated with Hag, Hag (delete flg22), and Hag premixed with 15 μM subtilomycin through luminescence assay. Standard errors were from six repeats. **c** Detection of ROS production in *A. thaliana* Col-0 treated with Hag and the complex Apn5 through luminescence assay. Standard errors were from six repeats. **d** Detection of Hag-induced ROS production in wild *A. thaliana* Col-0 leaf strips. The gradient concentrations of subtilomycin 7.5, 15, and 30 μM were applied to pre-incubate with Hag, respectively. The peak values were displayed. Standard errors were from five repeats. **e** Subtilomycin attenuates Hag-induced stomatal closure response. Stomatal response induced by different elicitors, Hag, Hag with subtilomycin, and flg22 (Ps) for 1 h in Col-0. The applied Hag and subtilomycin concentration is 30 and 15 μM, respectively. Error bars indicate SD. Two-sided Student's *T* test. **f** Quantitative RT-PCR assay on the expression of gene *frk1* after induction of Hag, flg22 (Ps), and Hag mixed with subtilomycin with gradient subtilomycin concentrations of 7.5, 15, and 30 μM. Standard deviations were calculated from three technical repeats. **g** Inhibition of Hag-induced *frk1::LUC* expression by subtilomycin in wild-type *A. thaliana* Col-0. Three independent biological repeats were applied. Two-sided Student's *T* test, *P* value = 0.005. Error bars were calculated from standard errors. The applied concentration of Hag and flg22 (Ps) is 30 and 1 μM, respectively

A recombination Hag with flg22 region replaced by flg22 (Ps), together with the previously mentioned truncated Hag with flg22 deleted (Fig. 3a), was employed to detect the interaction with subtilomycin. The MST results indicated that subtilomycin bound with both ΔHag (flg22) and Hag (ex) although with a minor decrease in binding (the ambiguous dissociation constant $K_d$ = 66.4 ± 9.11 and 80.6 ± 16.8 μM, respectively) compared with the former Hag ($K_d$ = 18.8 ± 4.15 μM) (Figs. 1f and 3h, i). Subtilomycin significantly attenuated the Hag (ex)-induced ROS and *frk1* gene expression (Fig. 3b, e). These results indicated that the depression effect led by subtilomycin was dependent on the interaction of subtilomycin with flagellin at the region beyond flg22 peptide.

**Subtilomycin enhances BSn5 endophytic colonization *in planta*.** To test whether subtilomycin and flagellin were expressed during BSn5 colonization in *planta*, the *yfp*-reporter strains, P$_{apnA}$-YFP and P$_{hag}$-YFP, had been respectively constructed to represent the expression of subtilomycin and flagellin. The *yfp* reporter strain P$_{tapA}$-YFP was constructed as a contrast to represent the expression of biofilm-related gene, which was previously proved essential for *B. subtilis* colonization in *Arabidopsis*[32] (Supplementary Fig. 11 and Methods). The confocal microscope imaging showed that all the genes were expressed *in planta*, although their expression patterns were different. Dispersive individual cells that express *hag* could be observed on/near the root surface from 6 to 36 h after inoculating. Cells in cluster could not express *hag* in 6 and 12 h, but they could express at a certain level in 24 and 36 h (Fig. 4a). P$_{apnA}$-YFP-positive cells tended to present in the manner of the root-attached cells cluster, but are very rare in the form of dispersive individual cells

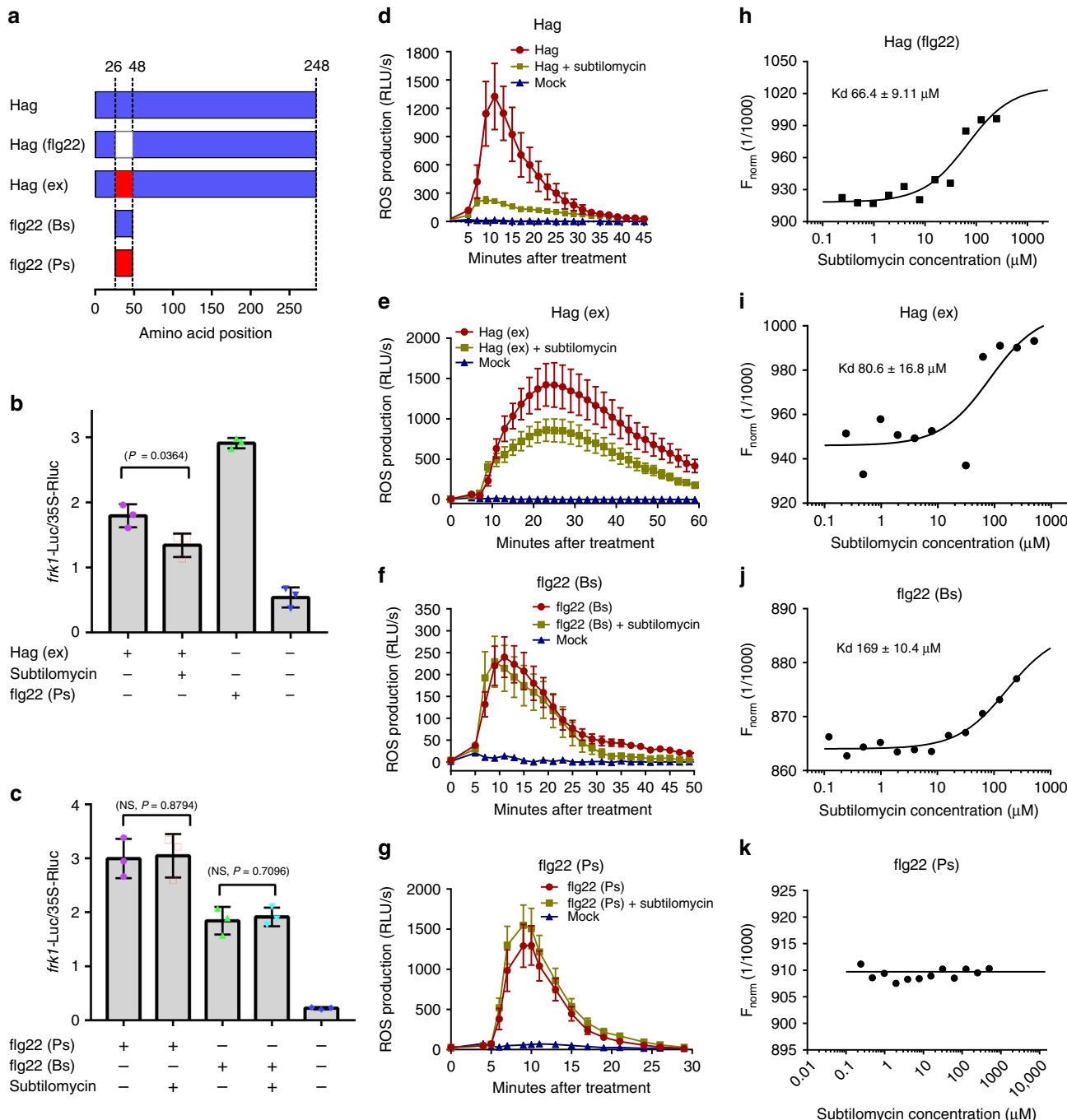

**Fig. 3** The suppression effects of subtilomycin on Hag-induced defensive response are dependent on binding the sites beyond flg22 region in Hag. **a** The diagram for constructing different recombination fragments of Hag. Detection of Hag (ex) (**b**), flg22 (Bs), and flg22 (Ps) (**c**) induced *frk1*-LUC activity in the presence or absence of subtilomycin, respectively. The sterilized ddH₂O was set as negative control. The standard deviations (s.d.) were from three technical repeats, and twice independent tests were applied with consistent results. NS refers to no significance. **d–g** Detection of the ROS production induced by Hag, Hag (ex), flg22 (Bs), and flg22 (Ps) in the presence or absence of 15 μM subtilomycin, respectively. For Hag, Hag (ex), and flg22 (Bs), $n =$ 12. For flg22 (Ps), $n = 6$. Error bars were from standard errors. The applied concentrations of Hag, Hag (ex), flg22 (Bs), and flg22 (Ps) are 60, 60, 30, and 1 μM, respectively. **h–k** MST assays on the interaction of subtilomycin with the peptides ΔHag (flg22), Hag (ex), flg22 (Bs), and flg22 (Ps), respectively

(Fig. 4a). The expression pattern of gene *tapA* was similar to gene *apnA*. These results indicated that gene *hag* was expressed *in planta*, which fits with the previous report, which claimed that gene *hag* was essential for *B. subtilis* chemotaxis and early colonization in *Arabidopsis*[33]. The expression of *hag* in cells cluster in 24 and 36 h might lead to plenty of flagellin existence in root circumstances and induce a plant defense. The expression level of gene *apnA* increased as the expression of gene *hag*, which

suggested that the interaction of subtilomycin and flagellin was reasonable in time and in space. The wild-type BSn5 without the YFP label was inoculated in the plant as a negative control. Just YFP-negative bacteria could be observed even 36 h after inoculating (Supplementary Fig. 12).

To characterize the endophytic colonization of BSn5 in *A. thaliana*, we employed two approaches. In the first approach, we employed an intensive Z-axis scanning by using the

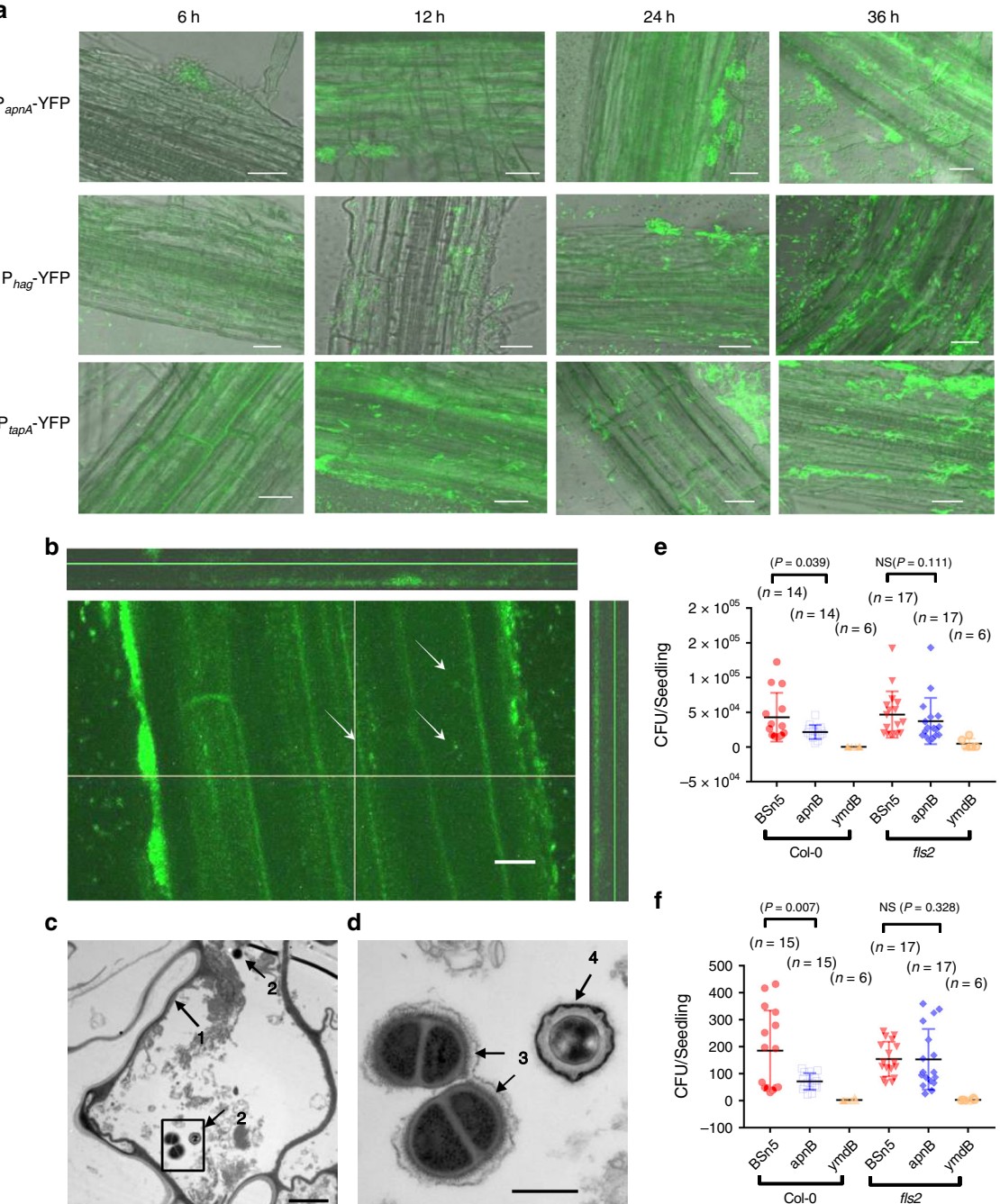

**Fig. 4 Subtilomycin production contributes to the colonization of *B. subtilis* BSn5 in *A. thaliana*. a** Microscope observation of the gene expression of subtilomycin (*apnA*), biofilm (*tapA*), and flagellin (*hag*) of BSn5 YFP-reporter strains in *A. thaliana*. The 6, 12, 24, and 36 h post inoculation were set for observation. Scare bars: 20 μm. **b** An intensive *Z*-axis scan was employed to observe the endophytic colonization of strain P*apnA*-YFP. When the layer below the root surface was scanned, some YFP-positive bacteria could be detected as the arrows indicated. Scare bars: 10 μm. **c** TEM observation of the cross-cutting section of *A. thaliana* root inoculated with bacteria BSn5 for 36 h. Scale bars: 2 μm. **d** Zooming of the squared area in picture **c**. Arrow 1 refers to the cell wall of plant cell. Arrow 2 refers to bacteria in intercellular space. Arrow 3 refers to the bacteria in division stage. Arrow 4 refers to spore. Scale bars: 500 nm. **e** Colonization assay of *B. subtilis* BSn5, subtilomycin mutant *apnB*, and biofilm mutant *ymdB* on the rhizoplane, and **f** endosphere of *A. thaliana* Col-0 and *fls2-1* seedlings, respectively. The independent sample *t* test was employed

confocal microscopy. From this test, when the layer far below the root surface was scanned, some bacteria with labeled *yfp* reporter were observed in the intercellular region (Fig. 4b). In the second approach, we observed the cross-cutting sections of root after inoculating BSn5 36 h under transmission electron microscopy (TEM). Some bacteria both in spores and in an active division stage were observed in the intercellular cave, although the amount is low as we expected (Fig. 4c, d).

These results proved that BSn5 could colonize inside *A. thaliana*.

To detect if the suppression action of subtilomycin to flagellin-induced defensive response finally benefits the producer colonization into plants, we employed BSn5 and its derivate strains including subtilomycin mutant *ΔapnB*, and biofilm-defect mutant *ΔymdB* (reported in *B. subtilis* NCIB 3610)[34], to inoculate wild-type *A. thaliana* Col-0 and flagellin-insensitive mutant *fls2-1*.

According to the acknowledged view, host defensive system was the main factor to drive the microbial community establishment in the range from rhizoplane to endosphere[2]. Thus, we investigated the colonization level of *B. subtilis* in rhizoplane and endosphere compartments, respectively. Here, the root surface washings were set as samples for rhizoplane. After root surface disinfecting and mashing, the homogenates of root were set as samples for endosphere (Supplementary Fig. 13, and Methods). From the results, wild-type subtilomycin producers had a stronger ability than subtilomycin mutant to colonize both on the surface and inside Col-0 seedlings. However, no significant differences were observed between subtilomycin-producing and mutant strains in rhizoplane or endosphere of *fls2-1* seedlings (Fig. 4e, f). The result means that the benefit that was provided by subtilomycin production for producer colonizing in plants was FLS2-dependent in *A. thaliana*. The biofilm mutant *ymdB* showed complete failure in colonization both in the rhizoplane and endosphere of *A. thaliana* (Fig. 4e, f).

**Subtilomycin producers showed correlation with plant origins**. The subtilomycin gene cluster was located on a prophage I region of BSn5 genome[35], which takes on as a hotspot with diversity among different strains in *B. subtilis* species (Supplementary Fig. 14). Through sequence blast, 17 potential subtilomycin producers with corresponding isolation sources were collected from the NCBI database (Supplementary Table 2). Nearly half of these strains (8/17) with available isolation information were isolated from plants, which suggested that the subtilomycin production in *B. subtilis* might relate to plants. BSn5 was initially isolated from elephant foot yam (*A. konjac*). We inoculated BSn5 and subtilomycin mutant in *A. konjac* seedlings to testify if subtilomycin also contributed to BSn5 colonization in *A. konjac*. From our results, subtilomycin significantly enhanced BSn5 colonization in *A. konjac* root (Fig. 5b). However, no significant difference was observed between the colonization ability of BSn5 and its mutant in rhizosphere soil (Fig. 5a). This result indicated that subtilomycin provided more benefits to producers to exist in root rather than in rhizosphere. Therefore, we investigated the relationship of subtilomycin production of *B. subtilis* isolates with their isolation origins including soil and plant environments.

We identified 93 *B. subtilis* group isolates from the samples from bulk soil and various grass plants (Supplementary Table 3). High-resolution LC–MS and PCR amplification showed that most *B. subtilis* subtilomycin producers (40/43) were isolates from plant environments, and few (3/43) were from free-living soils; accordingly, most non-subtilomycin strains (43/50) were isolates from soil, and few (7/50) were isolated from plants (Fig. 5c and Supplementary Table 3). This investigation suggested that subtilomycin was a strain-specific factor that was used by a broad range *B. subtilis* producers to adapt plant environments (Pearson chi-square value 57.753, $P < 0.001$).

## Discussion
Endophyte is an important composition of plant microbiota. However, non-symbiotic endophyte interactions with plant have been least studied. In this association, the microbes are usually horizontally recruited by plant each generation from soil. Therefore, it is a key for these microbes to overcome the plant defense and colonize in plant. Our results lead to a model in which endophyte *B. subtilis* reduces the stimulation of plant defensive response by producing lantibiotic subtilomycin to bind with self-produced flagellin, which makes subtilomycin producer a favorable colonist in plants (Fig. 6).

As the outcomes of commensal endophyte colonization in plant are beneficial, the selective pressures on host defensive system are different from pathogen. Through comparison of the flg22 sequences (Fig. 2a) and ROS assay on flg22 (Bs) and flg22 (Ps) (Supplementary Fig. 8), the beneficial *B. subtilis* retains a decreased potential to be sensed by plant defensive system. In our model, subtilomycin production will further reduce the stimulation to plant defensive response, which makes producers more favorable for plant endosphere colonization. From our results, a low level of colonization (100–300 cfu seedling$^{-1}$) of subtilomycin producers is observed in the endosphere (Fig. 4f). Nonetheless, the maintenance of a low-level existence in healthy plant endosphere for *B. subtilis* may be enough as an advantage in prior utilizing a dead plant, which fits the original and the high-frequency isolation of *B. subtilis* in dry grass well[36].

In most cases, the Zigzag model used to explain the pathogen–host association[18] may not fit well in the loose commensal association because pathogens are highly adapted to develop effectors to act on the specific targets in host cells and caused virulence effects. But from our model, the mechanism used by endophyte bacteria tended to be defensive. Fifteen micromolar subtilomycin that can significantly suppress the Hag-induced ROS cannot suppress the ROS induced by FliC (Se), a full-length flagellin cloned and expressed from *Salmonella enteritidis* (Supplementary Fig. 15). As subtilomycin binds to self-produced flagellin, this action mechanism does not harm the ability of plant defense to response on other pathogens. In addition, our mechanism is also different from the tight symbiotic bacteria–legume association[37]. In that relationship, high specific interaction factors are developed like a previous study mentioned. A bacterial BacA protein produced by symbiotic *Sinorhizobium* protected producers from the nodule-specific cysteine-rich antimicrobial peptides in plant[38].

As the FLS2 exists in most plants with reported sequences, our proposed mechanism may work as widely as the recognition of FLS2 to flagellin in plants. It explained why the *B. subtilis* subtilomycin producers have enhanced ability to colonize in both *A. thaliana* and *A. konjac*, which belong to far different taxonomies, and can be isolated from unrelated plants in our investigation (Supplementary Table 3).

Blast analysis with subtilomycin biosynthesis enzyme ApnB on the NCBI database, 85 analogs (with 29–37% identities) were discovered from the genomes of the *B. cereus* group and *Paenibacillus* genus (Supplementary Table 4). In Firmicutes, the bacteria from these taxa were most frequently isolated from plant endosphere niches[6]. According to the available origin records on the NCBI database, 55.3% (47/85) strains were from plant-related origins (Supplementary Tables 2 and Table 4). This analysis suggests that the mechanism may more generally be applied by endophyte bacteria to adapt endosphere niches.

Our work provides understandings to the association of beneficial endophyte bacteria with host plants, which have been studied least hitherto. This understanding provides a cue for the assembly of plant microbiota.

## Methods
**Bacteria strains and plants**. The bacterial strains used in this study included wild-type *Bacillus subtilis* BSn5 isolated from *A. konjac*[35], BSn5Δ*apnB* (subtilomycin deficiency, a subtilomycin biosynthetical enzyme for essential dehydration reaction was inactivated)[27], BSn5Δ*hag*, BSn5Δ*ymdB*, and *E. coli* strains BL21 and DH5α. *B. subtilis* strains were cultured in Luria-Bertani (LB) medium at 28 °C, and *Escherichia coli* was routinely cultured in LB medium at 37 °C. For selective media, the following antibiotics were added: ampicillin, 100 μg ml$^{-1}$ (*E. coli*); spectinomycin, 100 μg ml$^{-1}$ (*B. subtilis*); kanamycin, 20 μg ml$^{-1}$ for *B. subtilis* and 50 μg ml$^{-1}$ for *E. coli*. Plants used in this study included *Arabidopsis thaliana* Col-0, *fls2-1* mutant. *Arabidopsis* plants were grown in an illumination incubator AR800 (Ruihua, Chinese company) at 22/25 °C day/night and 70% relative humidity with a 10/14-h day/night light cycle for 4–5 weeks for leaf oxidative burst assay, stomatal response assay, or protoplast isolation. *Arabidopsis* plants used for bacteria colonization were planted in a Murashige and Skoog (MS) plate. After the seeds germinated in an MS

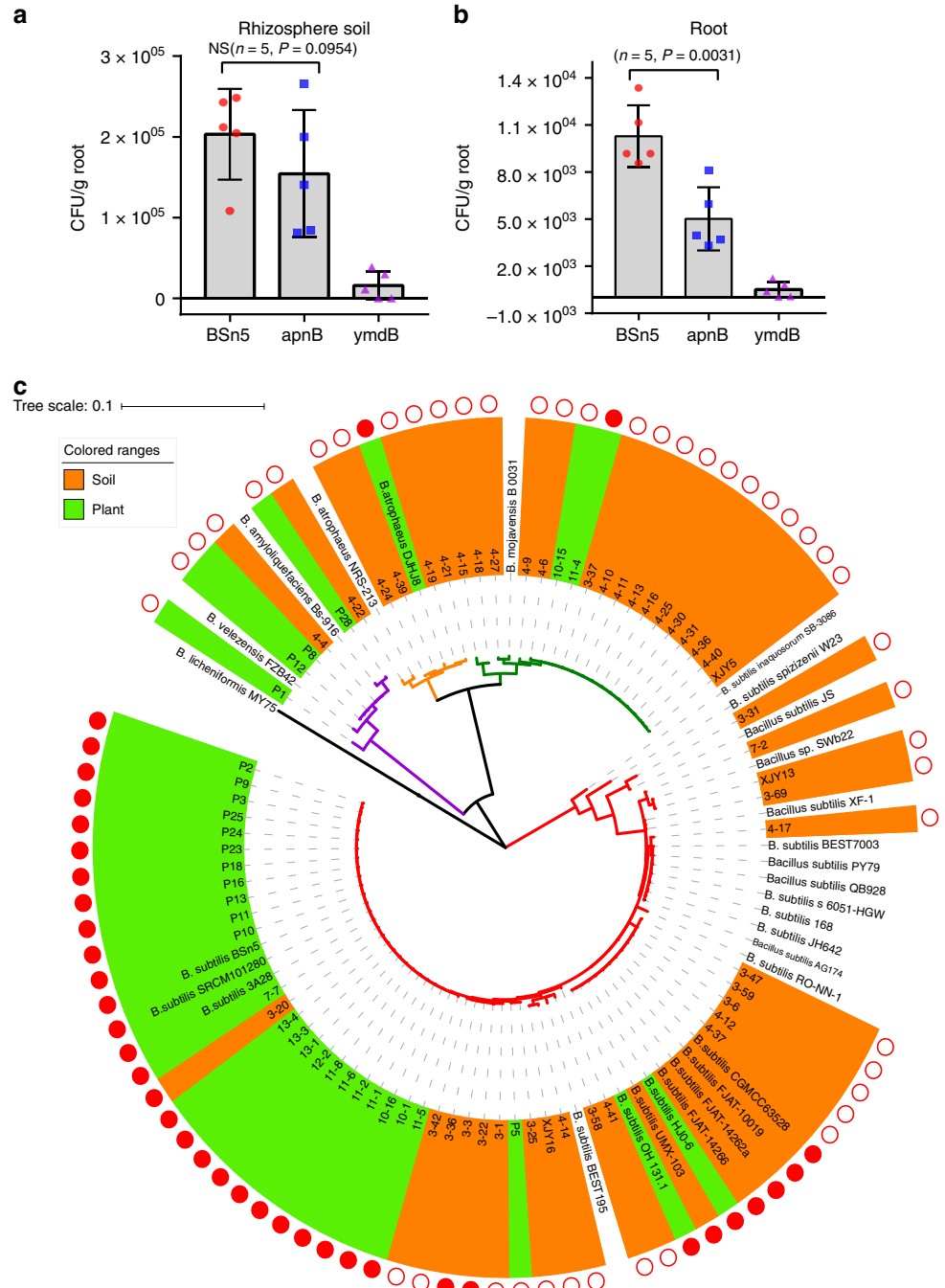

**Fig. 5** Subtilomycin production benefits for *Bacillus subtilis* to adapt on plants. Colonization assay *B. subtilis* BSn5, subtilomycin mutant *apnB*, and biofilm mutant *ymdB* in the **a** rhizosphere soil and **b** root of *A. konjac*. Error bars are s.d. from five replicates. **c** Phylogenetic analysis and basic features of 73 wild isolates and 29 NCBI strains of *B. subtilis* group based on the partial sequences in *gyrA*. Green background indicates isolates originated from plants. Orange background indicates isolates originated from soil. Solid red circle presents subtilomycin production. Hollowed red circle indicates non-subtilomycin production

plate, the seedlings were transferred into a 1/2 MS plate and cultivated for 7 days before inoculation bacteria in the same condition as mentioned above. *Arabidopsis* plants used for real-time quantitative PCR analysis were planted on a 1/2 MS plate with 1% sucrose and 0.6% agar for 10 days.

**Plasmid construction and *B. subtilis* transformation**. All plasmids in this study are listed in Supplementary Table 1. The construction of plasmids was based on standard techniques described by Sambrook and Russell[39]. Plasmids that were used for the expression and purification of Hag protein and its truncation fragments were constructed by PCR amplification of *hag* and the corresponding gene fragments into vector pMD18T (simple). These genes were cut by

PCR-induced sites *Bam*HI and *Xho*I and cloned into the vector pET28a. Plasmid that expressed truncation Hag with flg22 epitope deleted, Hag (del-flg22), was constructed by inverse PCR to amplify the pET28a-*hag*, which results in a PCR product with each ending *Eco*RI sites. The products were digested by *Eco*RI, and self-linked forming the pET28a-*hag* (del-flg22). Plasmid that expressed recombinant Hag with flg22 epitope replaced with flg22 from *Pseudomonas syringae*, Hag (ex), was constructed by PCR amplification of the flg22 (Ps) sequence with primers including homologous sequence ends with the linearized vector pET28a-ΔHag (flg22) digested by *Eco*RI and recombination with the linearized vector by using One Step Cloning Kit (Vazyme, Inc., China). All constructions were verified by sequencing and transformed into *E. coli* BL21 (DE3).

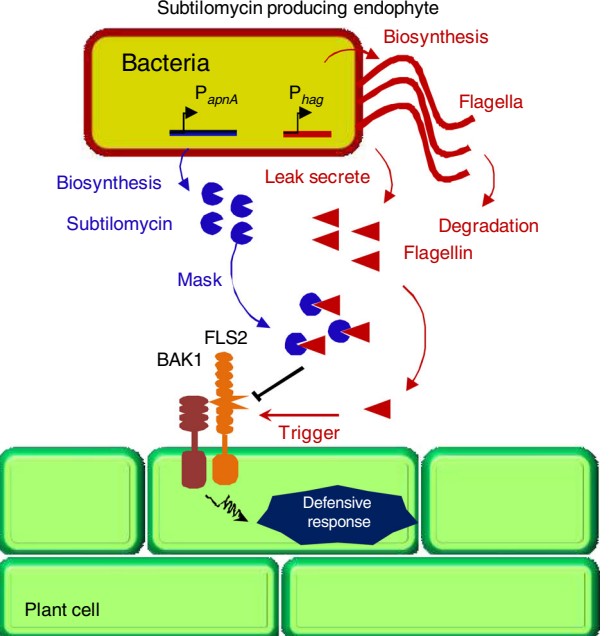

**Fig. 6** Model for masking flagellin by subtilomycin used by endophyte *B. subtilis* to evade plant defense. Subtilomycin producer uses subtilomycin to mask self-produced flagellin to evade plant defense. The red and blue arrows indicated the action of flagellin and subtilomycin, respectively

Gene inactivation plasmids were constructed as previously described in two ways[27]. The first was establishment of the *hag* mutant allele (*hag::spc*) by subcloning of the spectinomycin-resistance gene (*spc*) from pIC333 into the PCR-introduced *Cla*I and *Xba*I sites of the *hag* gene in pMD18T-*hag*, resulting in plasmid pB1206 (Supplementary Table 5). The second approach was the establishment of *ymdB* mutant alleles *ymdB::kan* by respectively amplifying the upstream and downstream arms of *ymdB* gene fragment into the upstream and downstream restriction sites of the integration vector pDG780, resulting in plasmid pB1208. Plasmids used for establishment of the flagellin, subtilomycin, and biofilm YFP-reporter strains were constructed by linking the *yfp* gene to the downstream of the promoter regions of subtilomycin structural gene *apnA* (encoding subtilomycin structural peptide), flagellin gene *hag*, and biofilm structure protein gene *tapA* (formerly, *tasA*). The constructed gene allele P*hag-yfp*, P*apnA-yfp*, and P*tapA-yfp* were cloned into vector pDG1730, resulting in pB1311, pB1312, and pB1313 for integration expression in BSn5.

The *B. subtilis* BSn5-competent cell preparation and transformation was performed by following the classical nutritional downshift method[40]. Five 10-min interval grades were set to capture the $t_0$ point (i.e., the point at which the culture leaves the logarithmic growth phase), considering the relative low-transformation efficiency of the wild-type strain. The constructed plasmids for gene inactivation and expression were used to transform forming recombinant strains. The vector pDG1730 harboring the corresponding genes was integrated into the *amyE* locus of BSn5 chromosome. Proper allelic replacement was confirmed by PCR and sequencing.

**Purification and detection of Apn5 by denature gel**. Preparation of the antibacterial protein Apn5 was carried out based on previously established procedures[27]. Briefly, the proper cultural time (12–16 h after inoculating) supernatant of BSn5 was salt precipitation by the saturation of 30% ammonium sulfate by centrifuging at 12000 r.p.m. at 4 °C for 40 min. After further dialyzing of the pellets, the soluble sample was saved as protein Apn5 and detected by denaturing 10% SDS–PAGE followed by the regular procedure. Apn5 protein was mixed with 5× loading buffer (250 mM Tris-HCl, pH = 6.8; 10% SDS; 0.5% bromophenol blue; 50% glycerol; 5% β-mercaptoethanol) and boiled for 5 min. After cooling, 20 μl of sample was loaded on the gel well.

**Native-PAGE assay of Apn5**. This assay was implemented by a precast-EZgel (gradient 8–20%) for the native-PAGE kit supplied by Solarbio (Solarbio LIFE SCIENCES, China) based on the manufacturer's manual. The electrophoresis condition is 150 V for 40 min at room temperature. Two parallel samples were loaded in the well. After the electrophoresis, the gel was cut into two parallel parts. One part was transferred into Coomassie brilliant blue stain and destain, the other part was washed with sterilizing ddH₂O for 10 min and used for in-gel in situ inhibition activity assay.

**Tricine–SDS–PAGE assay of Apn5**. To identify the antibacterial protein Apn5, Tricine–SDS–PAGE assay was performed based on the well-established protocol[41]. Sixteen percent gel was made based on the protocol. The sample Apn5 was mixed with 5× loading buffer and boiled for 5 min. Two parallel samples were loaded in the well. After the electrophoresis, the gel strips were cut and transferred on an indicator agar plate for inhibition assay.

**Size-exclusion chromatography**. The Apn5 protein was subjected to size-exclusion chromatography (Superdex-75 Increase 10/300; GE Healthcare). The technology support for SEC was provided by Yin P lab in Huazhong Agricultural University following the relative reference[42]. The buffer contained 25 mM Tris-HCl, pH 8.0, and 150 mM NaCl. The flow rate was set as 0.5 ml min⁻¹. All the fractions were collected as 0.5 ml per tube for further SDS–PAGE and inhibition activity assay. The relative amount of flagellin was determined by comparing the target protein band volume by using software Quantity One (Bio-Rad, USA). The relative amount of subtilomycin was determined by inhibition activity assay.

**Inhibition activity assay**. The classical agar well diffusion was used for inhibition activity assay as previously mentioned[27]. In brief, 20 ml of LB agar (1.0%, wt/vol) medium was inoculated (1/200, vol/vol) with indicator CU1065 suspension when it was cooled at 40 °C, and was poured into a 9-mm Petri dish. After drying for 15 min, proper wells were bored in each plate. Approximately 60 μl of active sample was loaded into each well. For in-gel in situ activity assay, the gel strips were put on the agar medium. The inhibition zone was observed and photographed after overnight culturing at 28 °C.

**Preparation of lantibiotic subtilomycin**. Purification of subtilomycin was carried out based on previously established procedures[27]. The crude subtilomycin was extracted from the Δ*hag* mutant derived from strain BSn5 that followed the same procedure for preparing Apn5. The target peak of subtilomycin was collected by loading the crude subtilomycin extracts in prep-HPLC. The collected target fragments were further transferred to rotoevaporation and lyophilization. The purified subtilomycin was saved at −70 °C.

**Protein expression and purification**. Protein Hag, ΔHag (flg22), and Hag (ex) were expressed by inducing the corresponding recombinant BL21 strains with 0.5 mM isopropyl-β-D-thiogalactoside when the cell density reached an OD₆₀₀ of 1.0. After growth at 37 °C for 3 h, the cells were collected and suspended with binding buffer (20 mM Tris, pH 8.0, 500 mM NaCl, and 10 mM imidazole). After further disruption by a high-pressure homogenizer (JNBIO, Inc., China), cell debris was removed by centrifugation for 30 min at 12,000*g* at 4 °C. The supernatant was collected and loaded on Ni²⁺-nitrilotriacetate affinity resin (Ni-NTA; CWBIO, Inc., China), followed by ten times column volume wash with 20 mM Tris, pH 8.0, 500 mM NaCl, and 50 mM imidazole. Elution was performed in buffer containing 20 mM Tris, pH 8.0, 500 mM NaCl, and 250 mM imidazole. The eluted proteins were transferred to Amicon 10-kDa cutoff Ultra-15 Centrifugal Filter Devices (Millipore) for buffer replacement with PBS buffer (pH 8.0).

**Immunoblotting analyses**. We performed Tricine–SDS–PAGE assay for subtilomycin. To detect the interaction of subtilomycin and Hag, a modified western blot method was applied. Purified subtilomycin was loaded on 16% (wt/vol) acrylamide tricine–SDS–PAGE gel. After electrophoresis, a copy of lanes was visualized by Coomassie brilliant blue staining. Another copy of lanes was applied for translating nitrocellulose (NC) membrane. After closing with 5% skim milk, the membrane was incubated with 1 mg ml⁻¹ His-tagged Hag for 2 h. After removing the free His-tag Hag, anti-his-tag antibody and secondary antibody linked to HRP were successively used for detection of His-tagged Hag according to the standard western blot produce.

**MST assay**. We used an MST assay to detect the interaction between subtilomycin and Hag as well as its variants. The procedure was followed according to the kit instructions that the manufacturer (Nano temper® Technologies) provided. Briefly, the method includes two steps, labeling 10 μM Hag protein with 30 μM fluorescent dye Cy5-NHS ester and preparing a serial two times dilution titration of the unlabeled molecule subtilomycin. Set the 1 mM subtilomycin as the highest concentration. Pretest the labeled proteins, and adjust the fluorescent response to 200–1000 by properly diluting with PBS buffer (pH 8.0). Mix the labeled proteins with the serial dilutions of subtilomycin very carefully, and load the samples on the model with standard capillaries. The scanning parameter is MST Power: 20% and LED Power: 60%. Fluorescence was measured by using a Monolith NT.115, and data were analyzed by using the supplied software NTAnalysis (Nano temper® Technologies).

**Oxidative burst**. The 4–6-week leaves were sliced into 1-mm strips and put into 100 μl of ddH₂O in a 96-well plate overnight. The luminescence assay was performed as previously described[43]. The reaction buffer contains 10 μg ml⁻¹ horse-radish peroxidase (HRP), 20 mM luminol, and MAMPs 1 μM flg22 (Ps); gradient concentrations flg22 (Bs) 60, 30, 15, 7.5, and 3.75 μM, 30 μM Hag, and its

recombinant fragments. To test the suppression of subtilomycin, 15 μM subtilomycin was mixed and incubated with related MAMPs for 10 min on ice before treatment. After removing the ddH$_2$O, the reaction mixtures were added into each well. Luminescence was recorded every minute for 50 min by a Tecan infinite 200 microplate reader.

**Stomatal assay**. The assay was performed as previously described[44]. Four- to five-week-old *Arabidopsis* were kept under light for 2 h to ensure that most stomata were opened before treatment. Leaf peels were carefully collected with tweezers from the abaxial side of mature leaves and put in deionized water buffer (25 mM MES, 10 mM KCl, pH 6.15), or buffer containing MAMPs on glass slides in square Petri dishes with lids on. The dishes were placed in the growth chamber that the plants were grown for 1 h before being observed under a light microscope. Images were randomly taken, and at least 30 stomata were recorded for each treatment. Stomatal apertures (length and width) were measured from these images with Adobe Photoshop.

**RNA isolation and RT-qPCR analysis**. Ten-day *Arabidopsis* seedlings were carefully collected from a 1/2 MS plate and transferred into 250 μl of sterile water in the 24-well plate (25 seedlings (~0.06 g) for each well) to float overnight. The proper concentrations of MAMPs were added to elicit the defensive response of the seedlings for 30, 90, and 180 min, respectively. After treatment, the seedlings were collected for RNA isolation. Total RNA was isolated according to the manufacturer's instructions by using a TransZol Up Plus RNA Kit (TransGen Biotech). The removal of DNA and reverse transcription were performed by using the PrimeScript RT reagent Kit with gDNA Eraser (TaKaRa) according to the manufacturer's instructions. RT-qPCR (real-time quantitative PCR) analysis was performed by The ViiA™ 7 system (Thermo Fisher Scientific) by using a Hieff™ qPCR SYBR Green Master Mix (Yeasen). The following PCR program was used: 95 °C for 30 s followed by 40 cycles of 95 °C for 5 s and 60 °C for 34 s. Fold change was calculated relative to mock (the plants treated with deionized water). Three experiments were used to calculate means and standard errors. The expression of gene FRK1 (AT2G19190) was normalized to that of the reference gene UBIQ10 (AT4G05320). The specific primer sequences are listed in Supplementary Table 6.

**Dual-reporter assay in protoplasts**. We followed the previously reported method[30]. Four- to six-week-old *Arabidopsis* plants were used for protoplast isolation. *Arabidopsis* protoplasts were prepared according to the Jen sheen's protocol[45]. The plasmids *frk1*::LUC (firefly luciferase) and *35S*::RLUC (Renilla luciferase), which were gifts from Prof. Jianmin Zhou, were prepared with the EasyPure HiPure Plasmid MaxiPrep Kit (TransGen Biotech) and co-transfected into *Arabidopsis* protoplasts with the ratio of 4:1, respectively. The protoplasts were incubated in the dark at room temperature for 18 h and treated with different MAMPs, including Hag, Hag (ex), and flg22 for 3 h. To study the suppression of subtilomycin, purified subtilomycin (final concentration is 15 μM) was premixed with another group of samples of Hag, Hag (ex), flg22 (Ps), and flg22 (Bs), respectively, and incubated on ice for 10 min before treatment. The added MAMPs were removed, and LUC and RLUC activity was measured by using the Dual-Luciferase Reporter system (Promega, E1910) according to the manufacturer's instructions.

**B. subtilis colonization assay in A. thaliana**. *B. subtilis* colonization assay in *A. thaliana* seedlings was performed and modified according to a previous description[46]. The 7-day seedlings of *A. thaliana* Col-0 and mutant *fls2-1* from a 1/2 MS plate were respectively soaked in LB containing 10$^6$ ml$^{-1}$ *B. subtilis* BSn5 and its derivative strains for 2 min. After another 7 days of culture in 1/2 MS plates, the inoculated plants were collected. Two seedlings were set as one sample. The seedlings were washed with 1 ml of saline, put into an Eppendorf tube containing 1 ml of saline and vortexed for 10 s, and then transferred into a new tube containing 1 ml of saline and vortexed for 30 s. The suspension was diluted with proper concentration for spreading the plate as a rhizosphere compartment. The seedlings were further transferred into a tube containing 3% sodium hypochlorite and made to stand for 2 min for surface disinfection. The remaining seedlings were washed with saline and ground by PowerMasher (Tiangen Biotech, Co., Ltd, China). The tissue homogenate was used for spreading the plate and counting as an endogenous compartment.

**Confocal microscopy**. The 7-day seedlings of *A. thaliana* Col-0 from a 1/2 MS plate were soaked in LB containing 2 × 10$^6$ ml$^{-1}$ *B. subtilis* BSn5 YFP-reporter strains, including P*hag-yfp*, P*apnA-yfp*, and P*tapA-yfp* for 2 min. The seedlings were placed in another 1/2 MS plate at 28 °C in the dark and were collected at 12 and 36 h for observation under confocal microscopy (Olympus FV1000). An intensive Z-axis scan was set for observing endophytic colonization. The scanning step is 0.24 μm, and 50 steps were set to scan. The total scanning depth was set >10 μm.

**TEM imaging**. The *A. thaliana* Col-0 seedlings inoculated with strain BSn5 for 36 h were chosen for making the sample for TEM observation. The leaf was

removed and the root with root hair zoom was cut into 10 mm. The tissues were transferred into fixing solution 2.5% glutaraldehyde for 2 h at room temperature. After washing with PBS (0.1 M) three times, 20 min each time, they were transferred into osmic acid for 2-h fixing. The tissues were washed again with the above conditions. The dehydration was employed to transfer the tissues through serious gradient ethanol solutions 30%, 50%, 70%, 80%, 85%, 90%, and 100%, 15 min for each gradient. The permeation was employed for the solutions in the order of acetone: epoxy as 2:1, 1:1, and 0:1, 12 h for each step, at 37 °C. The tissues were put into a capsule with epoxy and embedded for 48 h at 60 °C. The blocks were cut into proper size and shape for section. The section was made by a Leica ultramicrotome (EM UC7) with thickness 60–100 nm. After double staining with lead and uranium, the sections were observed under Tecnai G$^2$ 20 TWIN 200 kv TEM (FEI, USA).

**B. subtilis colonization assay in A. konjac**. The identical *A. konjac* seeds were selected for colonization assay. The plants were kept in square pots (7 × 7 cm), one plant per pot with nutrition soil in greenhouse at a consistent temperature of 28 °C. Ten milliliters of sterile water containing 2 × 10$^9$ cfu of *B. subtilis* strain P$_{tapA}$-YFP with Spc$^r$ label (standing for wild-type BSn5), mutant Δ*apnB* with Spc$^r$ label, and Δ*ymdB* with Kan$^r$ label were inoculated into the soil of 1-week generated *A. konjac* seedlings. Five plants were set for each treatment. Eighteen days after inoculation, the plants were manually harvested for investigation of *B. subtilis* colonization. The large soil aggregates were removed by shaking the roots. Since the root of one *A. konjac* plant is very big, parts of the roots are harvested and mixed for weighing. The collected roots with rhizosphere soil were put into the 50-ml tubes with 10 ml of sterile PBS. The tubes containing roots were sonicated three times 30 s cycle$^{-1}$. The washing buffer of the three times was subjected to centrifugation (1500 × *g*, 15 min). The pellets were set as rhizosphere soil sample. The roots were transferred into a new tube containing 10 ml of sterile PBS. The roots were sonicated for another three cycles 30 s cycle$^{-1}$. After the roots were put into another fresh 10-ml sterile PBS tube, the roots were ground by an electric pestle. The homogenate was set as root sample. The samples were used for spreading the plates with appropriate antibiotics after properly diluting. Colony-forming units were calculated. The cfu of rhizosphere soil samples and roots was normalized by using the corresponding root fresh weight with rhizosphere soil from each plant.

**Wild strains collection and isolation**. Soil samples were collected by removing the 5-cm surface soil from different areas of China (Supplementary Table 3). In all, 0.5 g of soil was transferred into 1.5-ml tubes, and was suspended with 1 ml of sterile ddH$_2$O, followed by 80 °C heating in a water bath for 10 min, and cooling at room temperature for 5 min. The supernatant was diluted to a proper concentration to spread the plate for observation of a single colony. Since the formation of remarkable biofilms of *B. subtilis* group strains, the colonies that exhibit obviously milky white, raised, dull, and wrinkled characteristics were selected for further identification based on *gyrA* gene sequencing. Plant samples were collected nearby in Wuhan city, Hubei province of China (Supplementary Table 3). The plants were carefully uprooted to avoid causing hurts. The collected samples were performed by the isolation procedure in a few hours, or saved at 4 °C for several days. The isolation procedure includes washing away the surface soil by using sterile water, dipping in the 75% ethanol for 5 min, grinding the plant tissues by using sterile mortar and pestle, and suspending in saline. Proper dilution was used for spreading the plate for observation of a single colony. Candidate bacteria were selected according to the above-mentioned observation.

**Detection of subtilomycin by LC–MS**. To verify if the wild *B. subtilis* isolates could produce subtilomycin, LC–MS (Agilent Q-TOF 6540) was used to detect subtilomycin from the 12-h culture supernatant according to the previously established method[27].

**Phylogenetic dendrograms**. Twenty-nine *gyrA* sequences of *B. subtilis* group strains were acquired from the NCBI genome database. Ninety-three *gyrA* sequences were acquired by sequencing the wild *B. subtilis* isolates. These DNA sequences were submitted to NCBI as a deposition (MN296126–MN296218). When the isolates from the same sample (listed in Supplementary Table 3) with the identical *gyrA* sequence, and the same subtilomycin production condition, which implies that they are probably repeat isolates, we will only retain one isolate to present in the phylogenetic analysis (Fig. 5b). Thus, 73 in 93 *gyrA* partial sequences of the wild isolates were displayed in the final phylogenetic dendrograms. The phylogenetic tree was constructed by using the Maximum Likelihood method based on the Tamura–Nei model[47]. There were a total of 853 positions in the final dataset. Evolutionary analyses were conducted in MEGA7[48]. The tree was drawn and annotated by the online software iTOL (http://itol.embl.de/)[49].

**Statistical analysis and reproducibility**. Sample sizes were determined based on pre-experiment or previous studies to reach a statistical significance (*P* < 0.05). Specifically, sample size for ROS assay, usually *n* ≥ 6, was chosen as they are common for experiments of that type[43]. Sample size for stomatal assay (*n* ≥ 30)[44], RT-qPCR, and LUC activity assay (*n* = 3) was chosen based on published methods[30]. The sample for colonization assay (*n* ≥ 5) was based on published *Bacillus*

colonization rates in *Arabidopsis*[46]. IBM SPSS statistics 20 software was used for statistical analysis. Two-sided Student's *T* test was used for statistical analysis of ROS assay, stomatal assay, RT-qPCR, and bacteria colonization assay. In most cases, biological replicates from independent samples were considered. Size-exclusion chromatography and MST assays were performed in duplicate (technical replicates).

**Reporting summary**. Further information on research design is available in the Nature Research Reporting Summary linked to this article.

## Data availability

Sequencing data can be accessed in NCBI by using accession numbers MN296126–MN296218. The source data underlying plots are shown in Supplementary Data 1. Data generated or analyzed in this study, which are not provided in the Supplementary Data, are available through request from the corresponding author. All materials can be obtained from the corresponding author.

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

## Acknowledgements

We thank Yangrong Cao for useful suggestions and Jian Liu and Ping Yin for providing technology for SEC assay, Shunping Yan for sharing the *Arabidopsis*

FLS2 seeds (College of Life Science and Technology, Huazhong Agricultural University), and Jianmin Zhou (Institute of Genetics and Developmental Biology, Chinese Academy of Science) for sharing the plasmids *frk1*-LUC and 35S-RLUC. We thank Pei Zhang and Anna Du (Wuhan Institute of Virology, Chinese Academy of Science) for TEM techniques. This work was supported by grants from the National Key R&D Program of China (2017YFD0201201); the National Natural Science Foundation of China (31600037); General Financial Grant from the China Postdoctoral Science Foundation (2016M600599); China 948 Program of Ministry of Agriculture (2016-X21).

## Author contributions

Y.D. designed, planned, performed, and analyzed experiments, and wrote the paper; H.C., C.L., J.X., Q.Q. and Y.X. performed the experiments; Y.Z., J.Z., D.P. and L.R. conceived and designed the experiments; M.S. designed the experiments and revised the paper.

## Competing interests

The authors declare no competing interests.
