## [Peer Review File · Communications Biology]

Reviewers' comments:

Reviewer #1 (Remarks to the Author):

The main claims of the manuscript is that the mechanism allowing *Bacillus subtilis* Bsn5 to escape plant defense for its endophytic colonization is the production of subtilomycin to mask flagellin responsible for activating plant defense.

The results are new and of interest to researchers working on plant-bacteria molecular dialogue, particularly those working on the endophytic behaviour of bacteria.

The work is convincing because of the complementarity of biochemical and genetic approaches, but I have two main concerns. They are both included in the title:

- Is the endophytic colonization of *A. thaliana* well characterized? Confocal observations and bacterial counting after "sterilization" of the root surface are not sufficient to prove this. Some bacteria embedded in their biofilm (cases of *B. subtilis*) may escape treatment with sodium hypochlorite. In Figure S9, two types of colonies are present in Photo 7: could you explain? Transmission electron microscopy is necessary to demonstrate that Bsn5 is an endophyte of *A. thaliana*.
- Is subtilomycin capable of specifically masking self-produced flagellin? This is important because otherwise it means that the subtilomycin of *B. subtilis* could help other bacterial species to escape plant defence. There is a lack of experience to evaluate the specific interaction between the subtilomycin of *B. subtilis* and its flagellin. For example, in Fig. 3d-g (but also in Fig. S5), the effect of subtilomycin on Bsn5's complete flagellin (Hag) is compared to its effect on *P. syringae*'s peptide flg22 (Fig. 3g-3k), not on *P. syringae*'s complete flagellin. However, it has been shown in Fig. 3f-3j that subtilomycin interacts weakly with Bsn5 flg22 (Fig. 3j) and does not inhibit the production of ROS by the plant (Fig. 3f).

The last two sentences of the conclusion are not adequate because bacterial endophytic behaviour cannot be considered as "ubiquitous", the beneficial effect of bacterial endophytism is not documented and its use as a biofertilizer is questionable.

The writing of the manuscript is not satisfactory because many errors remain (district_distinct_L106 ; purhing_washing_L212 ; soil_rhizosphere_L232 ; sere herbaceous?_L271...) and some presentation rules are not respected (e. g. space between values and units).

When figures include splicing it must be reported (Fig. S6b, S8a...).

Reviewer #2 (Remarks to the Author):

This article describes a mechanism used by endophyte *Bacillus subtilis* to mask one of its important MAMP, thus impeding its recognition by the plant defenses system and its entry in the plant. This is a novel and very interesting finding. Most of the results are abundant and convincing, many controls were used and the model was demonstrated using a good variety of techniques, which makes the manuscript scientifically very solid. However, the language needs much improvement, and certain conclusions have to be re-formulated because they do not correspond to the results presented.

Major comments:

The first part of the results lacks context (line 65 to 70). What is the *aiiA* transgenic breeding, is it relevant to the isolation of the strain? How was the antibacterial activity discovered? What is the proof that Apn5 has the antibacterial activity? The sum of experiments presented in Fig 1. are convincing for flagellin / subtilomycin binding, but they are presented poorly.

Figure 4A: Images are too small – we cannot visualize non-fluorescent bacteria. Thus, I cannot validate the author’s conclusions (e.g. positive hag-yfp cells not in contact with plant root at 36h, no tapA-yfp positive cells at 12h). This experiment do not lead to the conclusion that subtilomycin contribute to colonization. It just shows that it is expressed when cells are near the root.

Line 159-160: This conclusion is not accurate. It should read something like “the these results suggest that subtilomycin most likely binds flagellin outside the flg22 peptide, but this binding is required for the defensive response suppression.

Minor:

Line 36: What do you mean by « transforming nutrition » ?

Line 48-49 is unclear.

Line 168: What leads to the conclusion of “multiple sites”? It could be only one site.

Figure 3d, 3f and 3g need to be mentioned at line 157.

Reviewer #3 (Remarks to the Author):

In this manuscript, the authors claim that a bacterial isolate from *Amorphophallus konjac* is able to live as an endophyte more successfully by coating its flagellin with subtilomycin, which decreases the plant immune response. While the authors present a very interesting model, backed up by some in vitro work, the logic in this manuscript is difficult to follow, critical experimental controls are missing, and not all of the methods and results are fully explained, which make it extremely difficult for another researcher to reproduce the work. Particular aspects of the manuscript that could be improved with additional details and clarity include:

- 1) The main figure and the description in the results section do not fully demonstrate that Apn5 is solely composed of flagellin and subtilomycin. Is Figure 1A a non-denaturing gel, and Figure 1B a denaturing gel? If not, why are the components dissociating on Figure 1B? This is not described in the results or the methods sections. Does complementation with hag restore the phenotypes?
- 2) What is the difference between apnA and apnB? The authors never explain this, and they use different mutants in different experiments? Why?
- 3) For the experiment displayed in Figure 4, how long with the authors inoculate these seedlings? The methods do not describe this. Why did the author not normalize CFU by g of tissue? Did the seedlings vary in size?
- 4) For the experiment displayed in Figure 5, how do the authors know that there was not *Bacillus* present in the soil or vertically transmitted in their plants? Further, the different y-axes used in Figure 4b, 4c, 5a, and 5b make it difficult to interpret these findings.

Reviewers' comments:

Reviewer #1 (Remarks to the Author):

The main claims of the manuscript is that the mechanism allowing *Bacillus subtilis* BSn5 to escape plant defense for its endophytic colonization is the production of subtilomycin to mask flagellin responsible for activating plant defense.

The results are new and of interest to researchers working on plant-bacteria molecular dialogue, particularly those working on the endophytic behaviour of bacteria.

The work is convincing because of the complementarity of biochemical and genetic approaches, but I have two main concerns. They are both included in the title:

• Is the endophytic colonization of *A. thaliana* well characterized? Confocal observations and bacterial counting after "sterilization" of the root surface are not sufficient to prove this. Some bacteria embedded in their biofilm (cases of *B. subtilis*) may escape treatment with sodium hypochlorite. In Figure S9, two types of colonies are present in Photo 7: could you explain? Transmission electron microscopy is necessary to demonstrate that Bsn5 is an endophyte of *A. thaliana*.

Thanks for your comments and suggestions, which push us to improve our work. The concern of the reviewer about the endophytic colonization is important as we take it as a new claim that the lantibiotic subtilomycins not only enhance the colonization on root surface but also contribute advantages to the producer for endophytic colonization. Our responses to the reviewer are as follows:

1. We have tried two approaches to response the reviewer's concern about the endophytic colonization of the strain BSn5. In the first approach, we used TEM (Transmission electron microscopy) as the review suggested. Although the amount is low as we expected, some bacteria both in spores and in division stage were observed in the intercellular cave (please see Fig. 4c-d, the corresponding statement was added in manuscript line 180-184). In the second approach, we employed an intensive Z-axis scan using the confocal microscopy. From this test, when the layer below the root surface was scanned, some bacteria with labeled yfp were observed in the intercellular region (please see Fig. 4b, the corresponding statement was added in manuscript line 178-180). Both two manners could prove that BSn5 can colonize inside plant.

2. The most generally used definition on "endophyte" is bacteria or fungi that reside internally in plant tissues, can be isolated from the plant after surface disinfection, and cause no negative effects on plant growth (i.e., they are either beneficial or commensal)¹⁻². From the definition, we can see the surface disinfection is the most common way to distinct the surface bacteria from endophyte. Similar endosphere sample designation was also based on root treated by sonication³. The strain *B. subtilis* BSn5 was isolated from the callus during the construction of the *aiiA* gene transgenic *A. konjac*, which means BSn5 was isolated under a strict surface disinfection procedure. We believe that our disinfection treatment with sodium hypochlorite is intensive

enough for removing biofilm. Because if biofilm was remained, a bigger amount of bacterial cfu will be detected in the endosphere compartment rather than the low amount in our result (100-300 cfu/seedling, Fig. 4f).

3. It is a little different between endophytes from *Bacillus* and those well studied rhizobia and mycorrhizal fungi, which all form very obvious infection thread or symbiont organ. We thought endophyte from *Bacillus subtilis* live a facultative endogenous lifestyle and form loose association with plant. Slight amount of bacteria can finally enter into plant in an unclear mechanism, probably from open stomatal, wound, and root hairs by chance. We hold the views that the avoidance of plant defensive response will surely increase the chance for bacteria entering plant.

4. Many published references repeatedly showed bacteria from *Bacillus* class including *Bacillus subtilis* could be isolated and observed inside plant, including some recent results from culturomics and sequencing based approaches⁴⁻⁶. These researches prove *Bacillus subtilis* could exist in plant as endophyte.

5. Yes, like reviewer mentioned, there are two types of colony when we check the photo 7. We feel very conceived that one of them is *B. subtilis* that we inoculated in as the obvious white colony and biofilm wrinkles. Another one looks like strains from *Bacillus cereus* group as they are with waxy colony. We are not sure about the cause that the strains present in roots. Maybe it was caused by insufficient seeds disinfection or contamination during grinding operation. I must say that this kind contamination seldom happened in our tests, because we can tell the colony feather of *B. subtilis* well. Therefore, we think that this presentation does not substantially affect our optimization and determination of different surface disinfection conditions in this point. We set 2 replicates for testing this disinfection condition. Therefore, we choose to present another one for photo 7.

Is subtilomycin capable of specifically masking self-produced flagellin? This is important because otherwise it means that the subtilomycin of *B. subtilis* could help other bacterial species to escape plant defense. There is a lack of experience to evaluate the specific interaction between the subtilomycin of *B. subtilis* and its flagellin. For example, in Fig. 3d-g (but also in Fig. S5), the effect of subtilomycin on Bsn5's complete flagellin (Hag) is compared to its effect on *P. syringae*'s peptide flg22 (Fig. 3g-3k), not on *P. syringae*'s complete flagellin. However, it has been shown in Fig. 3f-3j that subtilomycin interacts weakly with Bsn5 flg22 (Fig. 3j) and does not inhibit the production of ROS by the plant (Fig. 3f).

We also paid attentions to the binding specificity problem like reviewer mentioned, and fully understood the reviewer's concern. In fact, we cloned a full-length flagellin encoding gene *fliC* from *P. syringae* into expression vector pET28a; however, the expression of the protein could only be detected in the forms of inclusive body in pellet of cell lysate. As alternative solution, we successfully cloned and expressed a full-length flagellin from *Salmonella enteritidis* (Please see Figure for communication). Through luminescence assay, 15 μ M subtilomycin that can significantly suppress the Hag induced ROS cannot suppress the ROS induced by FliC (Se) (Supplementary Fig. 11). This result proved the ROS induced by flagellin from *B. subtilis* was specifically suppressed by subtilomycin. The relative statement was added in discussion part.

Figure for communication. Expression of His-tagged FliC proteins from *Se* (*Salmonella enteritidis*) and *Ps* (*Pseudomonas syringae*) in *E. coli* strain BL21 detected by SDS-PAGE. Left picture shows cell lysate pellet and right picture shows the collected elution protein fragments by Ni⁺-NTA column purification.

We want to response the reviewer's worry about whether subtilomycin of *B. subtilis* could help other bacterial species to escape plant defense in another aspect. The expression of flagellin and subtilomycin are highly cooperated in time and in space, which was supported by the obtaining of the complex of subtilomycin and flagellin from the supernatant of BSn5. In addition, we want to communicate with the reviewer through two unpublished data, which also could support it. A self-produced protease AprE by *B. subtilis* could degrade subtilomycin quickly in our in-vitro assay (unpublished data), and subtilomycin could induce the expression of gene *hag* from our transcriptome data (unpublished data).

The last two sentences of the conclusion are not adequate because bacterial endophytic behavior cannot be considered as "ubiquitous", the beneficial effect of bacterial endophytism is not documented and its use as a biofertilizer is questionable.

Thank you for your suggestion. Yes, we agree with you. We deleted and revised relative sentences in the last paragraph.

The writing of the manuscript is not satisfactory because many errors remain (district_distinct_L106; purhing_washing_L212; soil_rhizosphere_L232; sere herbaceous?_L271...) and some presentation rules are not respected (e. g. space between values and units).

Thank you. We corrected all the mentioned errors by reviewer and highlighted these places with red font. We checked the space between values and units throughout the full text including texts, pictures, and tables, and corrected the presentation problems.

When figures include splicing it must be reported (Fig. S6b, S8a...).

Thank you. We added a report for each splicing picture and highlighted as red font.

Reviewer #2 (Remarks to the Author):

This article describes a mechanism used by endophyte *Bacillus subtilis* to mask one of its important MAMP, thus impeding its recognition by the plant defenses system and its entry in the plant. This is a novel and very interesting finding. Most of the results are abundant and convincing, many controls were used and the model was demonstrated using a good variety of techniques, which makes the manuscript scientifically very solid. However, the language needs much improvement, and certain conclusions have to be re-formulated because they do not correspond to the results presented.

We appreciate for the reviewer's positive comments and suggestions. We agree with the reviewer's criticism about the language and conclusions. The revised and supplemented places in manuscript were highlighted with red font.

Major comments:

The first part of the results lacks context (line 65 to 70). What is the *aiiA* transgenic breeding, is it relevant to the isolation of the strain? How was the antibacterial activity discovered? What is the proof that Apn5 has the antibacterial activity? The sum of experiments presented in Fig 1. are convincing for flagellin / subtilomycin binding, but they are presented poorly.

Thanks for reviewer's questions and kind suggestions. Our initial target is to find solutions for controlling the *A. konjac* soft rot diseases. For that, we transformed an *aiiA* gene, which encode protein can degrade the QS signals in pathogen of soft rot disease, from *Bacillus* into *A. konjac*, and constructed the transgenic *A. konjac* plant with good resistance to soft rot disease⁷. During the transgenic procedure, we isolated an endophyte *B. subtilis* from the callus culture of *A. konjac*. As our target is to control soft rot disease, we feel interested in the active compounds in the isolate; we expected to identify active compounds that can control plant disease, although it was finally proved that the active inhibition protein Apn5 was against *Bacillus* strain, not the pathogen of soft rot disease. However, it promotes the identification of the interaction between lantibiotic subtilomycin and flagellin and the proposal of the key hypothesis. We added some words in the appropriated place in the first results to state the initial think of our work with red font in the text.

A native PAGE assay has been supplemented to identify Apn5 as a complex of subtilomycin and flagellin in Fig. 1b. And the results description about the identification of Apn5 has been rewritten. We believe the part about the explanation and the presentation of Fig.1 has been improved.

Figure 4A: Images are too small – we cannot visualize non-fluorescent bacteria. Thus, I cannot validate the author's conclusions (e.g. positive hag-yfp cells not in contact with plant root at 36h, no tapA-yfp positive cells at 12h). This experiment do not lead to the conclusion that subtilomycin contribute to colonization. It just shows that it is expressed when cells are near the root.

Thank you for the reviewer's suggestions. We felt the conclusions in this part were not well

made. Therefore, we reformed the Fig. 4 and rewrote the description for it. Please check the changes in manuscript L165-L176.

Line 159-160: This conclusion is not accurate. It should read something like "the these results suggest that subtilomycin most likely binds flagellin outside the flg22 peptide, but this binding is required for the defensive response suppression."

Yes, thank you for your correction. We follow the reviewer's suggestion and rewrite the conclusion sentence as "These results suggest that the defensive response suppression of subtilomycin is more likely dependent on binding flagellin at the region outside the flg22 peptide."

Minor:

Line 36: What do you mean by "transforming nutrition" ?

Thank you for your suggestion. We have change "transforming nutrition" into "transferring some nutrients into available forms for plant (e.g. phosphorus and nitrogen)".

Line 48-49 is unclear.

Thank you. We rewrote this sentence as "The pathogen and host interactions are described as a Zigzag model⁸, which is like an arms race. In an arms race, usually one side would lose at the end."

Line 168: What leads to the conclusion of "multiple sites"? It could be only one site.

Thank you for your suggestion. We have rewritten the conclusion.

Figure 3d, 3f and 3g need to be mentioned at line 157.

Thank you, we added these fig notes into the appropriate place.

Reviewer #3 (Remarks to the Author):

In this manuscript, the authors claim that a bacterial isolate from *Amorphophallus konjac* is able to live as an endophyte more successfully by coating its flagellin with subtilomycin, which decreases the plant immune response. While the authors present a very interesting model, backed up by some in vitro work, the logic in this manuscript is difficult to follow, critical experimental controls are missing, and not all of the methods and results are fully explained, which make it extremely difficult for another researcher to reproduce the work. Particular aspects of the manuscript that could be improved with additional details and clarity include:

1) The main figure and the description in the results section do not fully demonstrate that Apn5 is solely composed of flagellin and subtilomycin. Is Figure 1A a non-denaturing gel, and Figure 1B a denaturing gel? If not, why are the components dissociating on Figure 1B? This is not described in the results or the methods sections. Does complementation with hag restore the phenotypes?

Thank you for your comment about the part for Apn5 identification. Fig. 1A is a denaturing SDS-PAGE gel following the regular procedure. From this denaturing gel assay, a main protein band with molecular weight about 30.0 kDa was presented, which indicated the sample was relative purified at least from the detection of SDS-PAGE assay. From the 2D SDS-PAGE-MS analysis, the most abundant protein spots 3-6 with scores >50 are identified as flagellin (Supplementary Fig. 1 and Supplementary Table 1). For this part, we added a native-PAGE assay data for Apn5 (Fig. 1b) and rewrote the results description for the Fig. 1a-d. We believe these adjustments could obviously improve the result presentation.

2) What is the difference between apnA and apnB? The authors never explain this, and they use different mutants in different experiments? Why?

Thank you for your comment. The gene *apnB* encodes a subtilomycin biosynthetic enzyme for essential dehydration reaction. This gene was inactivated for construction of subtilomycin mutant. The subtilomycin deficiency was identified in our previous publication⁹. The gene *apnA* encodes subtilomycin structural peptide. This gene was not used for construction of subtilomycin mutant here. Its promoter region was cloned into the upstream of YFP reporter vector to construct *PapnA*-YFP reporter strain for representing the expression of subtilomycin. The promoter activity of this region was identified in previous publication⁹. We added an explanation for each gene when it was mentioned in the first time and added descriptions in Method part with red font.

3) For the experiment displayed in Figure 4, how long with the authors inoculate these seedlings? The methods do not describe this. Why did the author not normalize CFU by g of tissue? Did the seedlings vary in size?

Thank you for your comment. We soaked the seedlings in diluted bacterial suspension (LB containing 10^6 mL⁻¹ *Bacillus subtilis* BSn5 and its derivative strains) for 2 min according to a previous method¹⁰. Considering the seedlings of *A. thaliana* are small and the seedlings are very identical in size by culturing on MS plate, we choose to use the CFU per seedling to represent the amount of colonization bacteria. Two seedlings were set as one sample. We added this description

in the method with red font.

4) For the experiment displayed in Figure 5, how do the authors know that there was not *Bacillus* present in the soil or vertically transmitted in their plants? Further, the different y-axes used in Figure 4b, 4c, 5a, and 5b make it difficult to interpret these findings.

Thank you for your comment. We indeed did not make this assay clear. We added some sentences about more details for this assay in red font in Method part. We used the recombination strain BSn5 (PtapA-YFP, *spc*^r) standing for wild type strain, mutant Δ *apnB* with the resistance gene *spc*^r label, and Δ *ymdB* with Kan^r label. The samples were used for spread the selective plates with appropriate antibiotics after properly diluting. Although there are still few natural antibiotic resistant bacteria could grow after 2-3 days in selective plate, our inoculated strains will grow faster and form a distinguishable colony after overnight, and we will count the cfu at the first time.

We made a mistake about the "CFU/seedling" in the Y-axis used in Fig. 5a. It should be "CFU/g root". It means the CFU in the definition rhizosphere soil part from a certain amount of root tissue, which has been weighted. Yes, the Y-axis used in Fig.4 and Fig. 5 are different, because the roots of *A. konjac* by culturing in pot with nutrition soil are much bigger and the sizes among individual plants are more different comparing with the *A. thaliana* seedlings that were planted in MS media plate. For the colonization assay in *A. konjac*, the CFU of rhizosphere soils sample and roots were normalized using the corresponding roots tissue fresh weight from each plant.

In addition, the purposes of these two assays are totally different. The colonization assay in *A. thaliana* would study if the subtilomycin contributed to the colonization of bacteria through regulating FLS2-flagellin defense. The colonization assay in *A. konjac* would study if subtilomycin contributed more for bacteria colonization in plant than in soil including rhizosphere soil. Considering *A. konjac* is the origin environment of strain BSn5, we chose them as a pair of host plant and subtilomycin producing bacteria in this assay, respectively.

Reference

- 1 Hallmann, J., Quadt-Hallmann, A., Mahaffee, W. F. & Kloepper, J. W. Bacterial endophytes in agricultural crops. *Canadian journal of microbiology* **43**, 895-914 (1997).
- 2 Coombs, J. T. & Franco, C. M. Isolation and identification of actinobacteria from surface-sterilized wheat roots. *Appl Environ Microbiol* **69**, 5603-5608 (2003).
- 3 Edwards, J. *et al.* Structure, variation, and assembly of the root-associated microbiomes of rice. *Proc Natl Acad Sci U S A* **112**, E911-920 (2015).
- 4 Bai, Y. *et al.* Functional overlap of the Arabidopsis leaf and root microbiota. *Nature* **528**, 364-369 (2015).
- 5 Huang, B., Lv, C., Zhuang, P., Zhang, H. & Fan, L. Endophytic colonisation of *Bacillus subtilis* in the roots of *Robinia pseudoacacia* L. *Plant Biol (Stuttg)* **13**, 925-931 (2011).
- 6 Gond, S. K., Bergen, M. S., Torres, M. S. & White, J. F., Jr. Endophytic *Bacillus* spp. produce antifungal lipopeptides and induce host defence gene expression in maize. *Microbiol Res* **172**, 79-87 (2015).
- 7 Ban, H. *et al.* Transgenic *Amorphophallus konjac* expressing synthesized acyl-homoserine lactonase (*aiiA*) gene exhibit enhanced resistance to soft rot disease. *Plant Cell Rep* **28**, 1847-1855 (2009).

- 8 Jones, J. D. & Dangl, J. L. The plant immune system. *Nature* **444**, 323-329 (2006).
- 9 Deng, Y. *et al.* Apnl, a Transmembrane Protein Responsible for Subtilomycin Immunity, Unveils a Novel Model for Lantibiotic Immunity. *Appl Environ Microbiol* **80**, 6303-6315 (2014).
- 10 Borriss, R. *et al.* Relationship of *Bacillus amyloliquefaciens* clades associated with strains DSM 7T and FZB42T: a proposal for *Bacillus amyloliquefaciens* subsp. *amyloliquefaciens* subsp. nov. and *Bacillus amyloliquefaciens* subsp. *plantarum* subsp. nov. based on complete genome sequence comparisons. *Int J Syst Evol Microbiol* **61**, 1786-1801 (2011).

REVIEWERS' COMMENTS:

Reviewer #1 (Remarks to the Author):

Numerous remaining errors (difficult to list all) and non-homogenous way to cite the references (inappropriate use of majuscule in some titles) must be corrected.

Some minor remarks

L18: subtilomycin bind with flagellin to mask the flg22 epitope: this is in contradiction with paragraph starting at L140 "Subtilomycin binds flagellin at the sites beyond flg22

L20: "...where the BSn5 was isolated..." instead of "...where the BSn5 is isolated..."

L39: "...plant defense..." instead of "...plants defense..."

L40: "...microbial endopytes..." instead of "...microbial invades..."

L44: "To avoid this defensive response" instead of "To overtake..."

L51: "misunderstood" instead of "blank"

L60: and many subtilomycin producers are found from various plants. Delete or to be separated from the first part of the sentence.

L64: Our initial target was to control instead of Our initial target is to control

L68: "we were interested" instead of "we feel interested"

L69-70: "of about 30 kDa..." instead of "which showed about 30 kDa"

L98: "exists" instead of "exist"

L216 & L219: "...samples from bulk soil..." instead of "...free living soil..."

L383: "2" h instead of "2 hours"

L393: space missing between pH and 8.0

L402: space missing between 1 and μM

L406: "min" instead of "minutes"

L406: "SpcR" instead of "spcR"

Fig. 4: "post-inoculation" instead of "past-inoculation". "on the rhizoplane" instead of "in the rhizoplane"

Reviewer #2 (Remarks to the Author):

The authors have answered all my comments.

REVIEWERS' COMMENTS:

Reviewer #1 (Remarks to the Author):

Numerous remaining errors (difficult to list all) and non-homogenous way to cite the references (inappropriate use of majuscule in some titles) must be corrected.

Some minor remarks

L18: subtilomycin bind with flagellin to mask the flg22 epitope: this is in contradiction with paragraph starting at L140 “Subtilomycin binds flagellin at the sites beyond flg22
Thank you. We have corrected the description in L 18 according to the review's suggestion.

L20: “…where the BSn5 was isolated…” instead of
“…where the BSn5 is isolated…”
Thank you. We have corrected it.

L39: “…plant defense…” instead of “…plants
defense…”
Thank you. We have corrected it.

L40: “…microbial endopytes…” instead of
“…microbial invades…”)
Thank you for your correction. However, we found our introduction did not reach to endophytes,
when we check the relative paragraph that reviewer mentioned. We have corrected the term
"invades" into a noun "invaders".

L44: “To avoid this defensive response” instead of “To
overtake…”
Thank you. We have corrected it.

L51: “misunderstood” instead of “blank”

L60: and many subtilomycin producers are found from various plants. Delete or to be separated
from the first part of the sentence.
Thank you. We have corrected it.

L64: Our initial target was to control instead of Our initial target is to control
Thank you. We have corrected it.

L68: “we were interested” instead of “we feel interested”
Thank you. We have corrected it.

L69-70: “of about 30 kDa…” instead of “which showed about
30 kDa”

Thank you. We have corrected it.

L98: exists; instead of exist;

Thank you. We have corrected it.

L216 & L219: samples from bulk soil; instead of free living soil;

Thank you. We have corrected it.

L383: 2; h instead of 2 hours;

Thank you. We have corrected it.

L393: space missing between pH and 8.0

Thank you. We have corrected it.

L402: space missing between 1 and M

Thank you. We have corrected it.

L406: min; instead of minutes;

Thank you. We have corrected it.

L406: SpcR; instead of spcR;

Thank you. We have corrected it.

Fig. 4: post-inoculation; instead of past-inoculation;

Thank you. We have corrected it.

. on the rhizoplane; instead of in the rhizoplane;

Thank you. We have corrected it.

Reviewer #2 (Remarks to the Author):

The authors have answered all my comments.

The authors have submitted an improved revised manuscript, which addresses the majority of the concerns brought up by the reviewers of the original manuscript version. However, there are a couple of clarifications and controls that would more fully resolve the story:

1) The addition of the non-denaturing gel in Figure 1B strengthens the story, but the gel image and visualization of the inhibition is not completely clear. Perhaps the addition of a negative or 0 Apn5 control would highlight the inhibition.

Thank you for your suggestion. We added a negative control in the native-PAGE assay in Figure 1B.

2) In Figure 2A, the colored arrows are not defined in the Figure legend or referred to in the text description.

We added a description for explaining the colored arrows.

3) For the YFP experiments displayed in Figure 4, fluorescence is present in each panel. Did the authors have a negative control to show YFP is not expressed for all promoters?

Thank you for your suggestion. Yes, we did. We added the observation of the wild type BSn5 strain as a non-YFP labeling negative control (Supplementary Fig. 9).